# Polyphenols, Autophagy and Neurodegenerative Diseases: A Review

**DOI:** 10.3390/biom13081196

**Published:** 2023-07-31

**Authors:** Vichitra Chandrasekaran, Tousif Ahmed Hediyal, Nikhilesh Anand, Pavan Heggadadevanakote Kendaganna, Vasavi Rakesh Gorantla, Arehally M. Mahalakshmi, Ruchika Kaul Ghanekar, Jian Yang, Meena Kishore Sakharkar, Saravana Babu Chidambaram

**Affiliations:** 1Department of Pharmacology, JSS College of Pharmacy, JSS Academy of Higher Education and Research, Mysuru 570015, India; cvichy97@gmail.com (V.C.); tousif.a.h7@gmail.com (T.A.H.); ammahalakshmi@jssuni.edu.in (A.M.M.); 2Center for Experimental Pharmacology and Toxicology, Central Animal Facility, JSS Academy of Higher Education and Research, Mysuru 570015, India; pavanhk@jssuni.edu.in; 3Department of Pharmacology, College of Medicine, American University of Antigua, Saint John’s P.O. Box W-1451, Antigua and Barbuda; nanand@auamed.net; 4Department of Anatomical Science, St. George’s University, West Indies FZ818, Grenada; gorantla55@gmail.com; 5Symbiosis Centre for Research and Innovation (SCRI), Symbiosis International (Deemed University), Pune 412115, India; director_scri@siu.edu.in; 6Drug Discovery and Development Research Group, College of Pharmacy and Nutrition, University of Saskatchewan, Saskatoon, SK S7N 5E5, Canada; jian.yang@usask.ca

**Keywords:** polyphenols, autophagy, neuroinflammation, proteinopathies, neurodegenerative diseases

## Abstract

Polyphenols are secondary metabolites from plant origin and are shown to possess a wide range of therapeutic benefits. They are also reported as regulators of autophagy, inflammation and neurodegeneration. The autophagy pathway is vital in degrading outdated organelles, proteins and other cellular wastes. The dysregulation of autophagy causes proteinopathies, mitochondrial dysfunction and neuroinflammation thereby contributing to neurodegeneration. Evidence reveals that polyphenols improve autophagy by clearing misfolded proteins in the neurons, suppress neuroinflammation and oxidative stress and also protect from neurodegeneration. This review is an attempt to summarize the mechanism of action of polyphenols in modulating autophagy and their involvement in pathways such as mTOR, AMPK, SIRT-1 and ERK. It is evident that polyphenols cause an increase in the levels of autophagic proteins such as beclin-1, microtubule-associated protein light chain (LC3 I and II), sirtuin 1 (SIRT1), etc. Although it is apparent that polyphenols regulate autophagy, the exact interaction of polyphenols with autophagy markers is not known. These data require further research and will be beneficial in supporting polyphenol supplementation as a potential alternative treatment for regulating autophagy in neurodegenerative diseases.

## 1. Introduction

Polyphenols are one of the major class of secondary metabolites found in edible and nonedible plants [1]. Polyphenolic compounds are classified based on the number of phenolic rings and functional groups such as phenolic acids (hydroxybenzoic acid and hydroxycinnamic acid), flavonoids (flavanols, flavonols, flavones, flavanones, isoflavones, proanthocyanidins), stilbenes and lignans [2]. The functional classification of polyphenols helps in understanding their bioavailability and health benefits. The main class of plant phenolic compounds are phenolic acids, also called phenolcarboxylic acids or polyphenols. They are divided into two types, hydroxybenzoic acid and hydroxycinnamic acid. Hydroxybenzoic acid has a complex structure and is present in low concentration in red fruits, black radish, onions, tea, etc. Hydroxycinnamic acid (HA) is present in higher concentration in the outer part of ripened fruits which includes caffeic acid, ferulic acid, p-coumaric and sinapic acid [3]. Phenolic acids are potent antioxidants and have several benefits such as being neuroprotective [4], antidiabetic [5], anticancer, anti-inflammatory, etc. [6]. Flavonoids are ubiquitously occurring polyphenolic compounds which are widely distributed in fruits and vegetables and have health benefits such as being antimicrobial, antioxidant, neuroprotective and having an antimalarial activity [7]. The classification of flavonoids is based on their benzopyran structure, the oxidation state of the carbon atom in the heterocycle ring. Flavonoids include flavanols, isoflavones, flavanones, flavonols, proanthocyanidins and flavones [8].

Anthocyanins are a major part of polyphenolic compounds. They are colored pigments that are dissolved in the vacuolar sap of epidermal tissues of plants. They are present in fruits, vegetables and flowers, for example, in grapes, apples, plums, berries, purple cabbages, red hibiscus, red rose, red pineapple sage, red clover, pink blossom, etc. [9]. Stilbene compounds are a larger group of natural-defense polyphenols in plants. They are found in low quantity in the human diet and the most well-known stilbene is resveratrol, which is present in red grapes, blueberries and wine and is reported to have antiaging and antiangiogenic activities [10]. Plant lignans are low-molecular-weight polyphenols that are present in rhizomes, leaves and roots and structurally possess two phenyl propene units. They are found richly in linseed as secoisolariciresinol and in minor quantities as matairesinol. These lignans are metabolized by the intestinal microflora to enterodiol and enterolactone. Lignans possess several pharmacological properties such as antitumor, antiviral and antiasthmatic properties [11].

Polyphenols are studied extensively for their health benefits to humans and animals. Polyphenols possess potent antioxidant and anti-inflammatory properties that help in the treatment of cancers, metabolic and neurodegenerative diseases [12,13], including age-related degenerative diseases [14,15]. Polyphenols provide neuroprotection and neurogenesis by regulating different signaling pathways, mitochondrial functions and antiapoptotic proteins. Polyphenols are capable of scavenging reactive oxygen species (ROS) directly and can thereby help regulate the mitochondrial apoptotic cascade. They exert an anti-inflammatory action by suppressing the expression of proinflammatory transcription factor NF-kB, which leads to apoptosis [16]. Polyphenols have been explored as nutraceuticals and for their usage in treating neurodegenerative diseases, and the majority of them have the ability to penetrate the blood–brain barrier [17]. Moreover, polyphenol supplementation reduces the incidence of psychiatric disorders as they possess an antioxidant activity [18].

Polyphenols such as curcumin and resveratrol produce a neuroprotective effect by activating the protein kinase signaling pathway and also via the upregulation of neurotrophic factors such as BDNF and GDNF [19]. Dysregulation in autophagy leads to neurodegenerative diseases, and evidence has revealed that polyphenol treatment ameliorates autophagy [20,21]. In this review, we summarize the absorption, distribution, metabolism and excretion (ADME), toxicity profile and effects of polyphenols on autophagy and proteinopathies in different neurodegenerative diseases. We used major search engines such as PubMed, SCOPUS, MDPI and Google Scholar to collect the scientific literature. To the best of our literature survey, we did not find many reports on the direct interaction of polyphenols with major proteins that regulate autophagy.

## 2. Absorption, Distribution, Metabolism and Excretion (ADME) of Polyphenols

In general, polyphenols have poor oral bioavailability due to their high metabolism, the interaction with the intestinal microflora and food matrices. Primarily, they are absorbed in the small intestine and metabolized by deconjugation reaction and deglycosylation. After absorption, they become less complex and undergo phase I (oxidation, reduction and hydrolysis) and phase II (conjugation) biotransformation in the enterocytes. This results in the formation of water-soluble conjugated metabolites (glucuronide, methyl and sulfate derivatives) that are released into the systemic circulation [22]. Polyphenols get excreted by two routes, urinary and biliary. More specifically, the biliary elimination of polyphenols occurs when the large and extensively conjugated metabolite is present, whereas in urine, small conjugates such as monosulfates are excreted [23].

The poor bioavailability of polyphenols is overcome by preparing nanoparticles with food macromolecules [24]. Liu et al. prepared and evaluated a curcumin nanodrug encapsulated in gelatin microspheres in a diabetic wound-healing model. These encapsulated microspheres increased the solubility and improved the bioavailability. Ravichandran et al., on the other hand, compared the kinetic profile of curcumin using conventional and nanoforms using an in vivo model. They found that the concentration of nanocurcumin in plasma and organs was increased in the nanoparticle-treated group (Figure 1) [25,26]. Similarly, Kohli et al. developed natural polysaccharide-based nanoparticles using berberine (BNPs) and found that the bioavailability of BNPs was increased in comparison to the normal free form in in vitro (drug release kinetics), ex vivo (gut permeation study) and in vivo models (Wistar rats) (Figure 2) [27]. Furthermore, the oral bioavailability of resveratrol was reported to be enhanced by preparing polymeric nanoparticles (in vitro and in vivo) [28]. Another interesting polyphenol is Epigallocatechin-3-gallate (EGCG). It is chemically unstable and has low bioavailability. Ramesh and Mandal prepared the solid lipid nanoparticles encapsulating EGCG, which improved the chemical stability and bioavailability [29]. In addition, the structural modification of EGCG by an acetylation process was shown to improve the bioavailability through in vitro and in vivo models [30]. Biasutto et al. added a glucosyl group by linking the succinate group to yield 3,4′,5-tri-(α-d-glucose-3-O-succinyl) resveratrol, which increased the pharmacokinetics when compared with conventional resveratrol in rats [31]. These experiments strongly suggest that novel drug delivery methods could potentially be explored further to improve the bioavailability of polyphenols.

## 3. Toxicity Profile of Polyphenols

Data reveal that some polyphenols exhibit a potential toxicity at high concentrations. For example, a high concentration of catechins was found to induce DNA damage in mice spleen cells [32], and grape extracts at 75 to 300 µg/mL concentrations promoted mitomycin C induced sister chromatid exchange in human peripheral blood lymphocytes [33]. A comparative toxicity study on quercetin, EGCG and cyanidin 3 glycoside revealed that quercetin caused a blood–brain barrier disruption and neurotoxicity [34]. Green tea polyphenols, when administered to mice at high doses, worsen colitis and colorectal cancer and cause kidney and liver dysfunctions [35]. Moreover, at higher doses, several of the polyphenols have been shown to promote mutagenesis or cause genotoxic and carcinogenic effects. This could potentially be attributed to their chemical structure, and the metabolites that are formed upon degradation. Hence, further research is needed to clearly establish the toxicity profile of polyphenols [36].

## 4. Autophagy and Neurodegenerative Diseases

Autophagy is a protective mechanism, as it eliminates outdated proteins, organelles and toxic substances and also reprocesses degraded products which is used as a source of energy in anabolic pathways [37]. Autophagy differs between eukaryotic cells and mammalian cells. In the case of eukaryotes, the proteins are degraded by two systems: the ubiquitin–proteasome system and macro-autophagy. In mammals, it is of three types: macro-autophagy, chaperone-mediated autophagy (CMA) and micro-autophagy. Mostly the degradation of misfolded proteins occurs through macro-autophagy in humans. Macro-autophagy executes in three phases known as autophagic flux: autophagosome formation, substrate recognition and autophagosome trafficking and degradation [38,39]. Stimuli such as cellular aging, pH, genetic mutations, oxidative stress resulting from excitotoxicity, native protein misfolding followed by immune activation and ionic strength alter the conformation of proteins [40,41]. The autophagy pathway is first regulated by phosphatidyl inositol 3-kinase (PI3K) along with ULK complex and initiates the formation of phagophore by expressing beclin-1 and AMPK and suppressing mTOR. Further, the formation of phagosomes is completed by two pathways, microtubule-associated protein 1 light chain 3 (LC3) and Atg5-12 [42]. Autophagosomes fuse with lysosomes to form autolysosomes which are responsible for clearing unfolded proteins. The misfolding of specific proteins and their subsequent aggregation cause proteotoxicity or proteinopathies in the brain. Dysregulation in any of the autophagy steps leads to the deposition of unfolded proteins or inclusion bodies in neurons and initiates the progression of neurodegenerative diseases such as Parkinson’s disease (α-synuclein), Alzheimer’s disease (Amyloid-β, neurofibrillary tangle), amyotrophic lateral sclerosis (SOD1, TDP 43) and Huntington’s disease (mutant Htt) [43,44] (Figure 3). Thus, the restoration of the autophagy mechanism is essential for treating neurodegenerative diseases. Several studies demonstrated the neuroprotective effects of polyphenols in neurodegenerative diseases (NDD), and in this review, we summarize the role of polyphenols in regulating autophagy in NDD.

### 4.1. Alzheimer’s Disease (AD)

Alzheimer’s disease is mainly characterized by the presence of amyloid-β plaques and neurofibrillary tangles in the brain and is responsible for memory loss. In AD, amyloid-β accumulation occurs outside the cell (extracellular), which is derived from an amyloid precursor protein through the sequential action of two proteases β and γ secretases [45]. The oligomers of amyloid-β peptide cause calcium dysregulation, mitochondrial dysfunction and inflammatory reactions. These processes promote the cytotoxicity of Aβ oligomers and the release of cytochrome c, triggering neurodegeneration [46]. In addition, the accumulation of phosphorylated tau proteins within neurons (intracellular) causes an axonal degradation and synaptic dysfunction [47].

Autophagy regulatory proteins such as beclin-1, PARK 2/parkin and nuclear receptor binding factor 2 (NRBF2) are dysregulated in Alzheimer’s disease [38]. The increased expression of microtubule-associated protein light chain-3 (LC3 I and II) and a significant decrease in Lamp-1 protein expression are recorded in the postmortem brain samples of AD patients [48]. In a transgenic mice model, the increased presence of aggregated autophagic vacuoles due to lysosomal dysfunction is reported [49]. Moreover, it was found that the dysregulation of the endosomal autophagic lysosomal pathway in the cortex and hippocampus is responsible for impairing the amyloid precursor protein processing [50].

The administration of polyphenol oleuropein aglycone has been shown to improve autophagic reactions and prevent or delay the occurrence of AD in CRND8 AD transgenic mice by increasing the expression of beclin-1, LC3-I and II, p62 [51] and by regulating the mammalian target of rapamycin (mTOR). Similarly, arctigenin, a polyphenol derived from *Arctium Slappa* (L.) is reported to inhibit the production of Aβ by suppressing β-site amyloid precursor protein cleavage enzyme 1 expression. This polyphenol also promotes the clearance of Aβ by enhancing autophagy through the AKT/mTOR signaling inhibition and the activation of the AMPK/Raptor pathway in an AD transgenic mouse model [52]. Additionally, resveratrol activates adenosine monophosphate activated protein kinase (AMPK) by increasing intracellular Ca^2+^ levels and inhibits mTOR, thereby regulating autophagy and the clearance of Aβ [53].

Lychee seed polyphenol (rutin, quercetin, etc.) inhibits the Aβ (1–42) induced activation of the NLR family pyrin domain containing 3 (NLRP3) inflammasome by acting through LRP1/AMPK-mediated autophagy in a transgenic mouse model [54]. An experiment conducted using resveratrol showed that resveratrol provided an anti-inflammatory effect against the Aβ-triggered microglial activation through TLR4/NF-jB/STAT cascade [55], both in in vitro and in vivo models.

Gintonin, a glycolipoprotein fraction isolated from ginseng, reported to decrease Aβ formation via nonamyloidogenic proteins in SHSY5Y cell lines. Gintonin then facilitated sAβPP release by increasing the [Ca^2+^] load and the activation of PI3K and PKC enzymes. Additionally, gintonin decreased brain neuropathy and memory impairment in wild-type Tg mice via the activation of the lysophosphatidic acid (LPA) receptor [56]. Parallelly, in a study using hippocampal neuronal cells, genistein was shown to protect the neurons from Aβ25-35-induced damage by acting through the estrogen-receptor-mediated pathway and antioxidant properties [57].

Evidence from the above studies suggests that polyphenols regulate autophagy through the AMPK–mTOR pathway by increasing the expression of beclin-1, LC3-I and LC3-II or through the LPA and estrogen receptor pathway in AD. Moreover, they protect from neuroinflammation by acting through the NLRP3 and TLR4/NF-jB/STAT cascade and thereby helps in the clearance of Aβ.

### 4.2. Parkinson’s Disease (PD)

The accumulation of α-synuclein (SNCA) in the form of Lewy bodies in the brain due to the overexpression of transcriptional and post-transcriptional mechanisms and a decreased degradation of α-synuclein through proteasomal and lysosomal dysfunction are the major pathological hallmark of PD [58]. Deubiquitinated SNCA is degraded by autophagy, and monoubiquitinated SNCA is removed by the proteasomal system [59]. The native SNCA protein is degraded by the CMA by proteases with the help of the HSC70 chaperone protein in the lysosomal membrane [60]. Apart from the degradation of SNCA, the autophagy pathway is also involved in mitochondrial turnover; therefore, the dysregulation of mitochondrial dynamics also results in Parkinson’s disease. Parkin, on the other hand, was found to facilitate mitophagy when impaired mitochondria undergo a macro-autophagy process [61,62].

In a study performed in A53T mutant and wild-type Drosophila models, isorhynchophylline induced mTOR-independent and beclin-1-dependent autophagy and stimulated the clearance of α-synuclein [63]. Similarly, curcumin was reported to reduce the accumulation of A53T α-synuclein by downregulating mTOR/p70S6k in SHSY5Y cells. Jiang et al. showed that curcumin produced neuroprotective effects in PD by facilitating the inhibition of inflammation, oxidative stress and α-synuclein aggregation through the mTOR-dependent autophagic pathway [64]. Resveratrol prevents rotenone-induced cell death by inducing heme oxygenase-1 expression and facilitating autophagy induction by promoting LC3-II and p62 levels [65]. Additionally, resveratrol was shown to enhance autophagic α-synuclein degradation by stimulating AMPK or via the activation of sirtuin-1 and LC-3 deacetylation in a 1-methyl-4-phenyl-1,2,3,6-tetrahydropyridine (MPTP) mouse model of PD [66]. In the MPTP model, gintonin, a glycolipoprotein, was reported to reduce α-synuclein accumulation in the substantia nigra and striatum of mice by regulating the Nrf2/HO-1 pathway [67]. Cell damage induced by overexpression of A53T α-synuclein was also reported to be reduced by caffeic acid by the activation of JNK/Bcl-2-mediated autophagy in SHSY-5Y cell lines and in transgenic mice models of PD [68]. These data indicate that a dysregulation in the autophagic pathway is improved by polyphenols through mTOR, AMPK, SIRT1, Ho-1 and JNK/Bcl-2 signaling, and polyphenols also increase the clearance of α synuclein and thereby help in the mitochondrial turnover in PD.

### 4.3. Huntington’s Disease (HD)

Huntington’s disease is an autosomal-dominant disease caused by the mutation in the HTT protein, and these mutated proteins form perinuclear cytoplasmic aggregates and intracellular inclusions which are normally removed through the autophagy process. The expanded polyglutamine tract (poly Q) present in the Huntington protein causes the protein to aggregate. The aggregated mutant protein is cytotoxic to specific neuronal population in the striatum and cortex and causes neurotoxicity [69]. In Huntington’s disease, the PIP3 binding protein linked with FYVE protein (Alfy/Wdfy3) is the adaptor protein required for the degradation of mutant HTT [70]. The dysregulation of macro-autophagy leads to HD pathogenesis, especially the ability of autophagic vacuoles to recognize cytosolic cargos [71]. In HD patients, the accumulation of mutant HTT recruits cytosolic beclin-1 resulting in the impairment of beclin-1 complex-mediated autophagy leading to neurotoxicity [72].

Neferine, a bisbenzylisoquinoline alkaloid from *Nelumbo nucifera*, is shown to alleviate HTT mutant proteins by inducing autophagy via an AMPK or mTOR-dependent manner in the PC-12 HD model [73]. Similarly, in a study carried out using an HEK293 cell model of HD, a phenolic extract from Murtilla berries was reported to reduce polyglutamine peptide aggregation and increase the expression levels of proteins p-62 and LC3-II [74]. Additionally, in an N171-82Q transgenic mice HD model, berberine was found to reduce mutant HTT by facilitating autophagy by increasing the conversion of LC3-I to LC3-II. Interestingly, these animals also show improved motor functions and prolonged survival [75].

Maher et al. observed that fisetin and resveratrol reduced the mutant HTT levels via the activation of extracellular signal regulated kinase (ERK) [76] in PC-12 cells, Drosophila (expressing mutant Httex1), and mice (R6/2). Moreover, resveratrol was shown protective against dopaminergic toxicity in Huntington’s by restoring ATG4, which is responsible for the formation of LC3-II using SHSY5Y cells [77].

Thus, it is clear that polyphenols have the ability to improve beclin-1-mediated autophagy in HD and potentially regulate autophagy proteins such as p62 and LC3 through the ERK, AMPK or mTOR pathways.

### 4.4. Amyotrophic Lateral Sclerosis (ALS)

ALS is a motor neuron disease characterized by a selective loss of the upper and lower neurons of the brain and spinal cord. The accumulation of superoxide dismutase-1 (SOD1) and transactive response (TAR) DNA-binding protein 43 (TDP-43) proteins due to a dysregulation in the autophagy process has been observed in patients with ALS and also in animal models [78]. Postmortem brain and spinal cord samples from ALS subjects have shown the presence of hyperphosphorylated, ubiquitinated fragments of TDP-43 protein [79]. In a clinical study, spinal cord motor neurons have shown increased levels of endoplasmic reticulum stress (ER-stress) and Er-stress-associated proteins [80]. Also in ALS, the aggregation of other proteins such as SOD-1, FUS and pNFH has been reported [81]. In a transgenic mice model of ALS, a small heat-shock protein B8 was found to reduce the aggregation of mutant SOD1 by increasing the solubility and clearance of mutant SOD-1 through an enhancement of autophagy without affecting the wild-type SOD 1 turnover [82].

In an in vitro model of ALS, resveratrol protects mutant-SOD-1-mediated toxicity by upregulating the expression of sirtuin-1 (SIRT-1) and motor neuron degeneration [83]. Several pieces of evidence show that resveratrol neuroprotection and the upregulation of LC3-II and beclin-1 by activating SIRT-1 and AMPK [84,85] improve ALS symptoms. A similar study using kaempferol from Brazilian green propolis produced protective effects in mutant SOD1 cell line models by increasing AMPK and inhibiting mTOR autophagy processes [86]. Dimethoxy curcumin, an analogue of curcumin, showed a protective effects in a cell-line model expressing TDP-43 mutant, by restoring mitochondrial damage, increasing the electron transfer chain complex, improving the transmembrane potential and upregulating uncoupling protein 2 (UCP2) [87]. The administration of fisetin to SOD1G93A mice and SOD1G85R Drosophila melanogaster increases the survival in mice by activating the extracellular signal-regulated kinase (ERK) pathway and decreases mutant SOD1 [88]. From the above data, it is evident that polyphenols interact with SIRT1, AMPK and ERK to facility autophagy and produce neuroprotection in ALS.

### 4.5. Multiple Sclerosis (MS)

Multiple sclerosis is an autoimmune disorder that involves neuronal loss, demyelination and chronic neuroinflammation [89]. Other than the clearance of proteins, autophagy plays an important role in both the innate and adaptive response. Autophagy plays a critical role in the regulating proinflammatory responses via interacting with NFκB. A blockade of autophagy increases the proinflammatory milieu [90]. Therefore, autophagy dysregulation prolongs the inflammatory response, which further results in autoimmune disease such as MS [91]. Blood samples obtained from MS patients revealed that numerous ATG genes were involved in the autophagy process [92]. The cerebrospinal fluid of MS patients showed upregulated levels of Atg5 and Parkin when compared with healthy subjects [93]. In chronic MS patients, an ultrastructural examination revealed the presence of synaptic vesicles containing autophagosomes in the dentate nucleus indicating the participative role of the autophagy process in MS pathogenesis [94].

From the pharmacological point of view, it is interesting to note that the administration of curcumin upregulated the protein expression of LC3-II and beclin-1 via the AKT/mTOR pathway, in turn shown to improve the motor function in a mouse model of experimental autoimmune encephalomyelitis (EAE). Further, curcumin downregulates the expression of inflammatory factors such as TNFα, IL-17, etc. [95]. Similarly, treatment with phloretin, a dihydrogen chalcone flavonoid, was shown to suppress neuroinflammation through AMPK-dependent autophagy in an EAE mouse model [96].

Matrine, a quinolizine alkaloid, was shown to upregulate the levels of beclin-1, LC3 and glutathione peroxidase and thereby to enhance mitochondrial autophagy in EAE [97]. Acteoside, a phenylpropanoid glycoside, obtained from *Rehmanniae radix*, was shown to suppress peroxynitrite (ONOO−) induced excessive mitophagy and to attenuate EAE [98]. These pathological and pharmacological pieces of evidence clearly reveal the role of polyphenols in autophagy regulation in MS.

The modulation of autophagy by polyphenols in neurodegenerative diseases (AD, PD, HD, ALS) has been summarized above and is diagrammatically shown in (Figure 4).

## 5. Limitations

Polyphenols as natural compounds are present in dietary foods and have many beneficial effects in the management or treatment of chronic diseases. Evidence from clinical and preclinical data suggests that polyphenols regulate autophagy. The exact interaction of polyphenols with autophagy protein markers still needs to be studied. This requires further research and will be beneficial in supporting polyphenol supplementation as a potential adjuvant in the management of neurodegenerative diseases. Moreover, some polyphenols possess some toxic effects and a low bioavailability; therefore, more preclinical and clinical studies are needed.

## 6. Conclusions

In this review, we summarized the role of polyphenols in regulating autophagy in various neurodegenerative diseases such as AD, PD, HD and ALS. Evidence from preclinical and clinical studies has revealed that polyphenols have the ability to regulate autophagy by modulating key autophagy proteins such as beclin-1, LC3 I and II and p62 and also act via the mTOR, AMPK, SIRT-1 and ERK pathways. Furthermore, evidence also shows that improving autophagy processes by polyphenols suppresses neuroinflammation and oxidative stress and improves the motor functions and memory in neurodegenerative diseases. However, more studies are warranted to support the use of polyphenols as supplements in modulating autophagy and proteinopathies and details on their interaction with autophagy proteins.

## Figures and Tables

**Figure 1 biomolecules-13-01196-f001:**
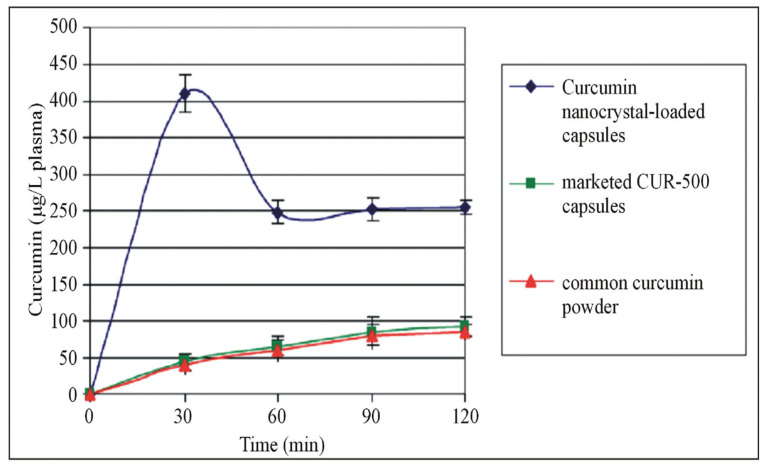
Pharmacokinetic profile of curcumin. Comparison of the pharmacokinetic profile of curcumin, nanocurcumin and marketed curcumin capsules. The figure is reused as per the journal’s copyright permission [26].

**Figure 2 biomolecules-13-01196-f002:**
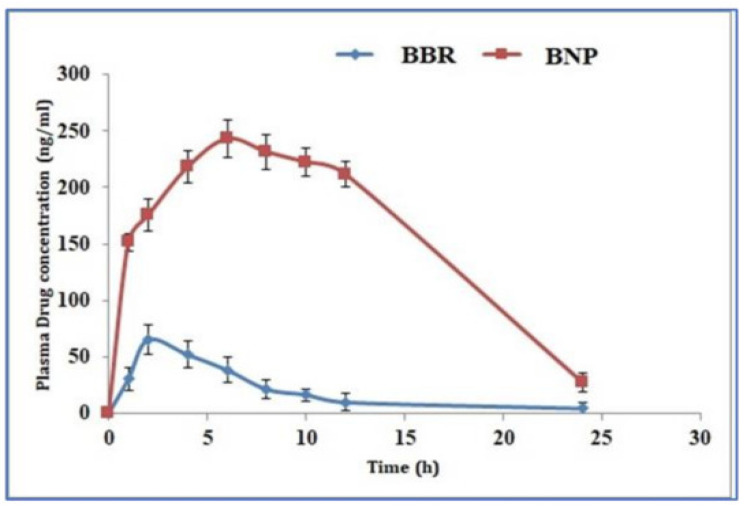
Pharmacokinetic profile of berberine. Comparison of pharmacokinetic profile of berberine and berberine nanoparticles. The figure is reused as per the journal’s copyright permission [27].

**Figure 3 biomolecules-13-01196-f003:**
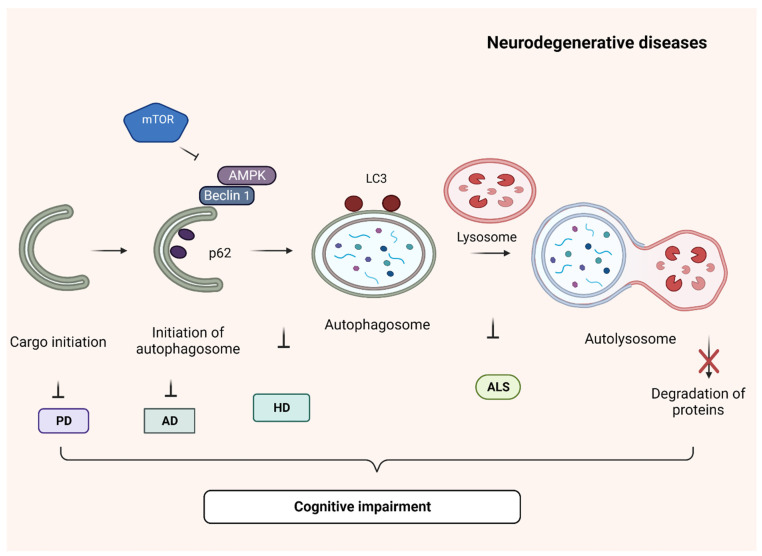
Autophagy dysregulation in different neurodegenerative diseases. Different stages of autophagy are affected in neurodegenerative diseases. Parkinson’s disease (PD)—the cargo initiation step, Alzheimer’s disease (AD)—initiation of autophagosome, Huntington’s disease (HD)—formation of autophagosome, amyotrophic lateral sclerosis—autolysosome formation is inhibited. These ultimately lead to cognitive impairment.

**Figure 4 biomolecules-13-01196-f004:**
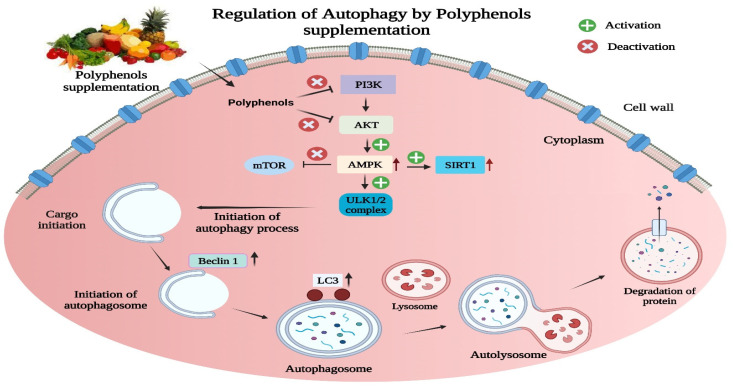
Regulation of autophagy by polyphenol supplementation in neurodegenerative diseases. Polyphenol supplementation regulates autophagy and aids in the clearance of misfolded proteins by entering cytoplasm. Polyphenols inhibit the PI3K/AKT pathway and thereby activate the AMPK and SIRT1 pathways and inhibit the mTOR pathway. Furthermore, they activate the ULK1/2 complex, which initiates the autophagy process by increasing the levels of beclin1 and LC3 proteins.

## Data Availability

The data that support the findings of this study are available in standard research databases such as PubMed, Science Direct or Google Scholar, and/or on public domains that can be searched with either keywords or DOI numbers.

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
