# Peer review of "Polyphenols, Autophagy and Neurodegenerative Diseases: A Review"

_biomolecules, 2023, doi:10.3390/biom13081196_

Round 1
Reviewer 1 Report
Biomolecules 2500452
Manuscript ID: Biomolecules 2500452
“Polyphenols, Autophagy and Neurodegenerative Diseases: A review” of Vichitra Chandrasekaran et al. explores the mechanism of action of polyphenols in modulating autophagy and summarize their role in critical pathways in particular for mTOR, AMPK, SIRT-1, and ERK, demonstrating how the polyphenols are able to increase in the levels of autophagic proteins. summarize
The work is original, well-structured and organized, and results are described in a relevant manner.
Taking into account the following considerations, in my opinion, the work can be accepted with major revision, as indicated below.
1. references
- Page 2, line 88, ref 18: the reference reported is not sufficient to confirm the statement. Please insert more convenient or add a further one.
- Page 2, line 95, paragraph 2.: In this paragraph we only mention the use of nanoparticles in order to improve the unfavorable pharmacokinetics of these polyphenolic molecules. The strategy of chemical modification is also used for this purpose. I suggest inserting a sentence to this research and optimization technique and one or two proper references as proposed or others: B.D. Arbo, C. Andre-Miral, R.G. Nasre-Nasser, L.E. Schimith, M.G. Santos, D. Costa-Silva, A.L. Muccillo-Baisch, M.A. Hort, Resveratrol derivatives as potential treatments for alzheimer's and Parkinson's disease, Front. Aging Neurosci. 12 (2020) 103-118; S. Intagliata, M.N. Modica, L.M. Santagati, L. Montenegro, Strategies to improve resveratrol systemic and topical bioavailability: an update, Antioxidants 8 (2019) 244; Marialuigia Fantacuzzi, Rosa Amoroso, Simone Carradori, Barbara De Filippis,
Resveratrol-based compounds and neurodegeneration: Recent insight in multitarget therapy, European Journal of Medicinal Chemistry, 233, 2022, 114242, ISSN 0223-5234, https://doi.org/10.1016/j.ejmech.2022.114242; Giacomini, E.; Rupiani, S.; Guidotti, L.; Recanatini, M.; Roberti, M. The use of stilbene scaffold in medicinal chemistry and multiTarget drug design. Curr. Med. Chem. 2016, 23, 2439–2489.
- Page 4, line 157, ref 35: is this reference adeguate?
2. Typos
- pay attention at the space between words (eg page 4, line 158 and 160) and put a full stop at the end of captions if required.
- not capital letter The, page 4, line 149
- correct Parkinson in line 241, page 6
3. in my opinion, it is necessary to explore also the autophagy in Multiple sclerosis and propose studies, if any, in this regard. This broadens the spectrum of neurodegenerative diseases.
Author Response
The authors thank the reviewers for their suggestions, we have made the corrections in the revised manuscript
Reviewer 2 Report
In this paper, the authors focused on polyphenols and their relationship with many autophagy-related pathways, whose dysregulation seems to among the leading causes underlying several neurodegenerative diseases. In the first section, they described the chemical and functional classification of polyphenols, mentioning the main ones and their pharmacokinetics. Then, they explained the process of autophagy itself, whereas in the last part they dealt with the interactions between polyphenols and autophagy within the most common and disabling neurodegenerative disorders, i.e., Parkinson’s disease, Alzheimer’s dementia, Huntington’s chorea, and amyotrophic lateral sclerosis. Although many papers on the effect of polyphenols on neurodegenerative diseases are available, the authors reviewed this relevant and timely topic in an original modality. In particular, they highlighted the involvement of polyphenols in some pathways (such as mTOR, AMPK, SIRT-1, and ERK) and concluded that polyphenols cause an increase in the levels of autophagic proteins like beclin-1, microtubule associated protein light chain, and Sirtuin1, among others. Overall, the review is nicely conceived and introduced, comprehensively carried out, and supported by several bibliographic references. Few comments to the authors, requiring minor revision.
- Abstract: please further mention the clinical implications of the findings emerging from this review.
Introduction: the role of beneficial dietary regimens (e.g., PMID: 36907474), as well as of some bioactive compounds in terms of potential neuroprotective effects (especially for cognition) and risk reduction towards different psychiatric disorders (such as geriatric depression) should be further emphasized, as recently demonstrated by several publications (e.g., PMID: 36986159 and PMID: 36771380, respectively).
- Conclusions: before this section, a brief paragraph listing the main limitations and caveats would be useful.
- Line 51: replace “has” with “have”; line 180: “…causes memory loss” with “responsible for memory loss”.
Minor editing of English language required.
Author Response
As per the comments given by the reviewer we have made the corrections in the revised manuscript

Round 2
Reviewer 1 Report
considering the modifications done to the manuscript and the accurate adherence to the suggested indications, the work can be accepted for publication